# Surface Plasmon-Enhanced Luminescence of CdSe/CdS Quantum Dots Film Based on Au Nanoshell Arrays

**DOI:** 10.3390/ma12030362

**Published:** 2019-01-24

**Authors:** Chun-Li Luo, Rui-Xia Yang, Wei-Guo Yan, Chun-Mei Chen, Shu-Yu Liu, Shi-Jin Zhao, Wen-Qi Ge, Zhi-Feng Liu, Guo-Zhi Jia

**Affiliations:** 1School of Electronic and Information Engineering, Hebei University of Technology, Tianjin 300400, China; luocl@tcu.edu.cn (C.-L.L.); 201621901009@stu.hebut.edu.cn (C.-M.C.); 2School of Control and Mechanical Engineering, Tianjin Chengjian University, Tianjin 300384, China; zhaosj0309@163.com (S.-J.Z.); gewenqi@tcu.edu.cn (W.-Q.G.); 3School of Science, Tianjin Chengjian University, Tianjin 300384, China; 18722366058@163.com (S.-Y.L.); dip-coating@163.com (G.-Z.J.); 4School of Materials Science and Engineering, Tianjin Chengjian University, Tianjin 300384, China; tjulzf@163.com

**Keywords:** quantum dots, enhanced luminescence, surface plasmon resonance, Au nanoshell structures

## Abstract

In this paper, Au nanoshell arrays, serving as a photo-activated material, are fabricated via the combination of self-assembled nanosphere lithography and the thermal decomposing polymer method. The intensity and position of surface plasmonic resonance can be tuned from the visible region to the near-infrared region by changing the size of Au nanoshell arrays. When resonance absorption peaks of metal nanoparticles are matched with emission wavelengths of core-shell CdSe/CdS quantum dots, fluorescent intensity of CdSe/CdS quantum dots can be strongly enhanced. The physical mechanism of fluorescent enhancement is explained.

## 1. Introduction

Quantum dots (QDs), as a low-cost and multifunctional material, have excellent optoelectronic properties, such as narrow emission band width and tunable emission in the full visible spectral range. They have been widely concerned and applied in optoelectronic devices and light-emitting diodes [1,2,3,4] for several decades. Recently, the luminescence of QDs film has attracted more and more attention of researchers due to special optical properties. Some experiments found that the luminescence intensity of QDs could be enhanced due to the interaction between QDs with metal nanoparticles [5,6,7]. The physical mechanisms of fluorescence enhancement are thought of as the surface plasmon resonance effect of metal nanoparticles. In particular, when emission wavelengths of QDs are matched with resonance absorption peaks of metal nanoparticles, the fluorescence of QDs films will be enhanced dramatically. The intensity and position of the surface plasmon resonance can be modulated through changing the size, shape, and distribution of metal nanoparticles. Thus, various metal nanoparticles and two-dimensional plasmonic arrays have been designed to enhance the photoluminescence intensity and quantum yield of the QDs in the last decade. For example, some works mainly focused on enhancing the luminous efficiency of QDs in solution environments by means of the interaction between Au/Ag nanoparticles and QDs [8,9,10,11]. In addition, some scholars focused on improving the luminous efficiency of QDs film based on metal nanostructure arrays [12,13,14,15,16,17,18,19,20,21]. However, luminous properties of core-shell CdSe/CdS QDs film based on metal nanostructure arrays have been little investigated. 

In this study, Au nanoshell (NS) arrays are used to enhance fluorescence properties of core-shell CdSe/CdS QDs. These Au NS arrays are successfully fabricated based on polystyrene spheres (PS) arrays combined with thermal decomposition by electron beam deposition. The size of Au NS arrays can be controlled by choosing different PS spheres. Surface plasmon resonance of Au NS arrays may be modulated from the visible to near infrared region by adjusting the size of Au NS arrays. In addition, fluorescence enhancement mechanisms of core-shell CdSe/CdS QDs film based on Au NS arrays are further explained. 

## 2. Experimental

### 2.1. Fabrication of PS Arrays with Different Layers

The self-assembly of PS with different sizes were performed according to the same procedures as those in our previous work [22,23,24]. Briefly, the monolayer colloidal PS arrays, with different diameters (500, 750, and 1000 nm), were self-assembled on clean quartz substrates by the interface self-assembly method. Figure 1a–e shows the schematic illustration of the fabrication of the colloidal PS arrays. Figure 1a is the monolayer PS arrays, which was fabricated by interface self-assembly. The samples were placed in an air-drying oven and the monolayer colloidal PS arrays were solidified on the substrate at 60 °C for 60 min. Figure 1b is the process of fabricating the second layer of the PS arrays. The second and third layers of PS arrays were prepared through repeating the above step, as is showed in Figure 1c–e. 

### 2.2. Fabrication of Au Nanoshell Arrays and Core-Shell QDs Coated on Au Nanoshell Arrays

The 20-nm thick gold film was deposited on PS arrays by the electron beam deposition method. Figure 1i–k shows the monolayer, two layers, and three layers of the PS arrays coated with Au shell arrays. Due to the instability of the PS sphere, all samples were annealed at 300 °C in a resistance heating furnace for 1 h to remove PS spheres. When the PS spheres decomposed, Au shell arrays form on the quartz. The scanning electron microscopy (SEM) of Au nanoshell arrays was measured after the Au nanoshell arrays were pasted on tape. From SEM of figure supporting-information Appendix A, we could confirm that PS spheres were removed completely. To make core-shell CdSe/CdS QDs film coat uniformly on Au shell arrays, all samples were placed in an O_2_ plasma cleaning chamber for 30 s plasma treatment, with the oxygen flow rate of 200 mL/min and a radio-frequency power of 70 W. Then, 20 microlitre core-shell CdSe/CdS QDs solution was dropped on the surface of Au NS arrays, which showed a strong hydrophilic interaction after O_2_ plasma cleaning. Thus, the core-shell CdSe/CdS solution formed a uniform layer on Au NS arrays, as shown in Figure 1l–n. 

### 2.3. Characterization

The morphology and composition of the samples were investigated using a field-emission SEM (JEOL Ltd., Tokyo, Japan). A UV-visible spectrophotometer (Varian Cary 300, Shanghai Sunny Hengping scientific Instrument Co., Shanghai, China) was used to characterize optical properties of QDs coated on Au NPs. Photoluminescence (PL) measurements were performed by the excitation from a 325-nm line of a continuous-wave He–Cd laser to analyze the efficiency of fluorescent spectra.

## 3. Results and Discussion

### 3.1. Morphologies of Au Film Coated on PS Arrays

Figure 2a,c,e show the SEM images of 500, 750, and 1000 nm PS arrays, respectively, coated with 20-nm thick Au film on the substrate by an interface self-assembly and transferred method. From the SEM images, it was obviously seen that the PS arrays kept their hexagonal distribution during the interface self-assembly process, according to the principle of minimum energy. This interface self-assembly and transferred method provided us an effective and simple way to fabricate two and three layers of PS arrays. When PS arrays were removed by annealing in a resistance heating furnace, the morphologies of the Au nano-shells still remained in the hexagonal arrangement. After 20 microlitre core-shell CdSe/CdS QDs solution drops on the surface of Au NS arrays, it was obviously seen that the morphologies of Au nano-shell arrays coated with CdSe/CdS QDs became blurred, from the Figure 2b,d,f, due to the poor electric conductibility of CdSe/CdS QDs. In addition, it was difficult to differentiate single CdSe/CdS QDs by SEM due to the small size. 

Figure 3 shows SEM of 500 nm PS arrays with different layers. From the SEM of Figure 3a, it was clearly seen that PS arrays were arranged orderly with a closed honeycomb structure. To confirm the as-prepared PS arrays with three layers, the 30° cross-sectional views of 500 nm PS arrays with different layers were investigated using SEM. Figure 3b shows that 500 nm PS spheres formed a monolayer ordered self-assembled structure, from the 30° cross-sectional view. The second layer of the PS arrays were transferred onto the first layer of the PS arrays by the interface self-assembly method. From Figure 3c, we could obviously see that the PS arrays uniformly covered the surface of the first layer of the PS arrays. The 30° cross-sectional SEM of the third layer of the PS arrays, in Figure 3d, showed that the PS arrays still remained ordered structures. So, the PS arrays with controlled layers could successfully be prepared by the interface self-assembly method.

### 3.2. Optical Properties of Au Shell Arrays with Different PS Layers

To contrast the influences of Au film on PS arrays, the absorption spectra of Au nanoshell structures with PS spheres, and those without PS spheres, were tested by UV-visible spectrophotometer. For different layers of PS arrays coated with Au shell arrays, we could obviously see that the main absorption valleys occurred at about 650 nm and secondary absorption valleys occurred at about 790 nm, as shown in Figure 4a. In addition, these valleys showed a weak red-shift with the increase of the number of layers. In the absorption spectra, the intensity of absorbance peaks with monolayer PS arrays was stronger than those of other layers, as shown in Figure 4a. Thus, we can conclude that the number of PS array layers has almost no effect on the position of absorption peaks. A small variation of the intensity of absorption peaks with layers could be caused by the order of arrangement between layers.

To obtain stable Au NS arrays, all samples were annealed at 300 °C in a resistance heating furnace for 1 h. Due to the thermal instability of PS spheres, PS spheres decomposed at 300 °C and Au film coated on PS arrays formed Au shell arrays. Optical properties of Au NS arrays were further detected by the UV-visible spectrophotometer. From the results, we could see that absorption peaks of the Au NS arrays were significantly different from those of Au film coated on PS arrays, as is shown in Figure 4b. The main absorption valley of Au NS arrays occurred at 500 nm and the secondary absorption valley occurred at 720 nm. It is deduced that PS spheres were decomposed by annealing at 300 °C for 1 h by the variety of absorption spectra before and after annealing. In addition, we can observe Au NS arrays from the SEM in Appendix A. When PS spheres were removed, Au film coated on PS sphere arrays attached to the substrate and formed Au NS arrays. From the spectrum of Figure 4b, it can be seen that the position of the absorption peaks of the Au NS arrays, prepared with multilayer PS arrays, and the monolayer PS arrays were almost the same. However, the intensity of absorption peaks of Au NS arrays prepared with the monolayer PS arrays was superior to others.

### 3.3. Tunable Optical Properties of Au Shell Arrays with Different Diameter PS Arrays

To achieve tunable optical properties of Au shell arrays, different diameter PS spheres were used to prepare Au NS arrays. Figure 5 shows the absorbance spectra of Au shell arrays with the diameters of 750 and 1000 nm. We could obviously see that the main absorbance valleys of both the 750 and 1000 nm Au NS arrays were all located at 500 nm, or nearby, which was consistent with the 500 nm Au NS arrays. Therefore, it could be deduced that the resonance valleys of Au NS arrays was determined by the intrinsic properties of the Au material. For 1000 nm Au NS arrays, there were two absorbance peaks, located at 700 and 900 nm, respectively. Secondary absorbance peaks of 750 nm Au NS arrays were located at 780 nm, or nearby. It was not hard to find that absorbance peaks of 750 and 1000 nm Au NS arrays, as they were obviously red-shifted compared with those of 500 nm Au NS arrays. The above results indicated that the absorbance peaks were determined by the period of the Au shell arrays. Therefore, tunable resonance peaks of Au shell arrays could be regulated by choosing appropriate PS arrays. 

To compare the effects of the influence of different PS arrays on Au NS arrays, the optical properties of Au NS arrays, prepared by monolayer PS arrays, were discussed in detail. The resonance absorption peaks of 500, 750, and 1000 nm Au NS arrays red-shifted from 590 to 800 nm with the increase of the Au shell size, as is shown in Figure 6. For 500 nm Au NS arrays, the resonance peak was located at 600 nm, which was consistent with the emission peak of CdSe/CdS QDs. The resonance peak of 750 nm Au NS arrays is located at 780 nm, or nearby, and we deduce that of 1000 nm Au NS arrays is located at the near infrared region, which were mismatched with the emission peak of CdSe/CdS QDs. That is to say, 750 and 1000 nm Au NS arrays could not provide more energy to CdSe/CdS QDs film compared with 500 nm Au NS arrays. We deduced that the fluorescence signal of CdSe/CdS QDs coated on 500 nm Au NS arrays may be enhanced effectively based on the surface plasmonic resonance effect. 

### 3.4. Enhanced Fluorescence of CdSe/CdS QDs Based on Au NS Arrays 

To verify the above assumptions, the room temperature PL spectra of bare CdSe/CdS QDs film and CdSe/CdS QDs coated on Au NS arrays, with the NS diameters of 500, 750, and 1000 nm, respectively, are detected by the fluorescence spectrum. From the PL spectra in Figure 7, we could clearly see that the emission peak of bare CdSe/CdS QDs was located at about 570 nm. However, the luminescence intensity of CdSe/CdS QDs film was very weak. The main reason is that CdSe/CdS QDs film coated on the substrate was very thin, and even sparse, and thus resulted in the low efficiency of luminescence. To enhance the luminescence efficiency of CdSe/CdS QDs, CdSe/CdS QDs film coated on Au NS arrays were prepared by monolayer and multilayer PS arrays. The resonance absorption peaks of Au NS arrays can be tuned by adjusting the size of the PS sphere. When the absorption peak of Au NS arrays matches with the emission peak of the CdSe/CdS QDs, surface plasmon resonance produced by Au NS arrays would more effectively enhance the luminescence efficiency of CdSe/CdS QDs. From the results, as shown in Figure 7d, it can be seen that the PL intensity of CdSe/CdS QDs coated on 1000 nm Au NS arrays was nearly as much as bare CdSe/CdS QDs film. That is to say, the surface plasmon resonance peak based on 1000 nm Au NS arrays mismatched with the emission peak of CdSe/CdS QDs. Therefore, Au NS arrays prepared by monolayer and multilayer PS sphere arrays also do not bring about more energy to CdSe/CdS QDs. For CdSe/CdS QDs coated on 750 nm Au NS arrays, the luminescence of CdSe/CdS QDs was more significantly improved than 1000 nm Au NS arrays. That is to say, the 750 nm Au NS arrays can provide more energy to CdSe/CdS QDs film than 1000 nm Au NS arrays. From the absorption spectra, we could obviously see that the intensity of the absorption peak of 750 nm Au NS arrays, at 570 nm or nearby, was stronger than 1000 nm Au NS arrays. As far as 500 nm Au NS arrays are concerned, the luminescence intensity of CdSe/CdS QDs film was demonstrably better than the others. The luminescence intensity increased by about quintuple, compared with bare CdSe/CdS QDs film. At the same time, it was not difficult to find, as shown in Figure 6, that the intensity of the absorption peak of 500 nm Au NS arrays was stronger than other Au NS arrays. Therefore, the intensity and position of the absorption peak of Au NS arrays determines the luminescence efficiency of CdSe/CdS QDs film. In addition, we calculates the relative fluorescence quantum efficiency of QDs based on monolayer Au nanoshell arrays according to the procedure [25]:
$$Y_{u}=Y_{s}\times (F_{u}/F_{s})\times (A_{s}/A_{u})\times (n^{2}{}_{u}/n^{2}{}_{s})$$
$Y_{u}$ and $Y_{s}$ represent the fluorescence quantum yields of the measured substance and the reference substance, respectively; $F_{u}$ and $F_{s}$ represent the integral area of fluorescence emission spectra corresponding to the measured substance and the reference substance under the excitation wavelength, respectively; $A_{s}$ and $A_{u}$ indicate the absorbance of the reference substance and the measured substance at the excitation wavelength, respectively; and $n_{u}$ and $n_{s}$ indicate the refractive index of the measured substance and the reference substance, respectively. For QDs film based on 500 nm monolayer Au nanoshell arrays, fluorescence quantum efficiency reaches up to 8.3%, and the QDs film based on 750 nm and 1000 nm are 6.4% and 3.2%, respectively.

## 4. Conclusions

In this paper, we demonstrate luminescence enhancement of CdSe/CdS QDs film assisted by surface plasmon resonance on Au NS arrays. Au NS arrays are fabricated via the combination of self-assembled nanosphere lithography and the thermal decomposing polymer method. The results show that Au NS arrays prepared from single-layer PS could largely enhance the luminescence efficiency of CdSe/CdS QDs film; superior to those of the two-layer and three-layer PS arrays. In addition, the luminescence intensity could be enhanced more efficiently when the resonance peaks of Au NS arrays are matched with the luminescence peaks of QDs. Therefore, Au NS arrays act as an auxiliary layer to enhance the luminescence efficiency of different QDs film. 

## Figures and Tables

**Figure 1 materials-12-00362-f001:**
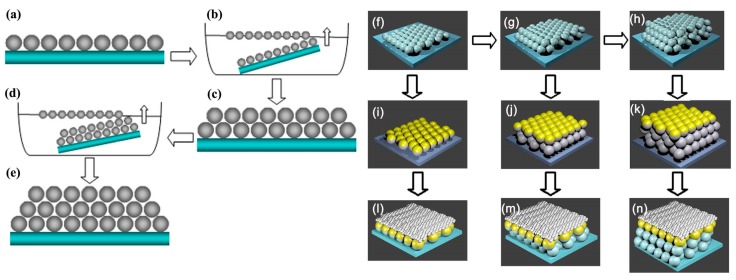
The schematic illustration of the process: (**a**) The monolayer colloidal PS arrays; (**b**) fabricating process of the two layers of the PS arrays by interface self-assembly; (**c**) the two layers of the PS arrays on the substrate; (**d**) fabricating process of the three layers of the PS arrays by interface self-assembly; (**e**) three layers of the PS arrays on the substrate; (**f**–**h**) monolayer PS arrays, double-layer PS arrays and triple-layer PS arrays; (**i**–**k**) a 20 nm Au film on the monolayer, two layers, and three layers of the colloidal PS arrays; (**l**–**n**) the Au shell arrays after decomposing the PS spheres; (**j**–**l**) CdSe/CdS QDs film coating on Au shell arrays.

**Figure 2 materials-12-00362-f002:**
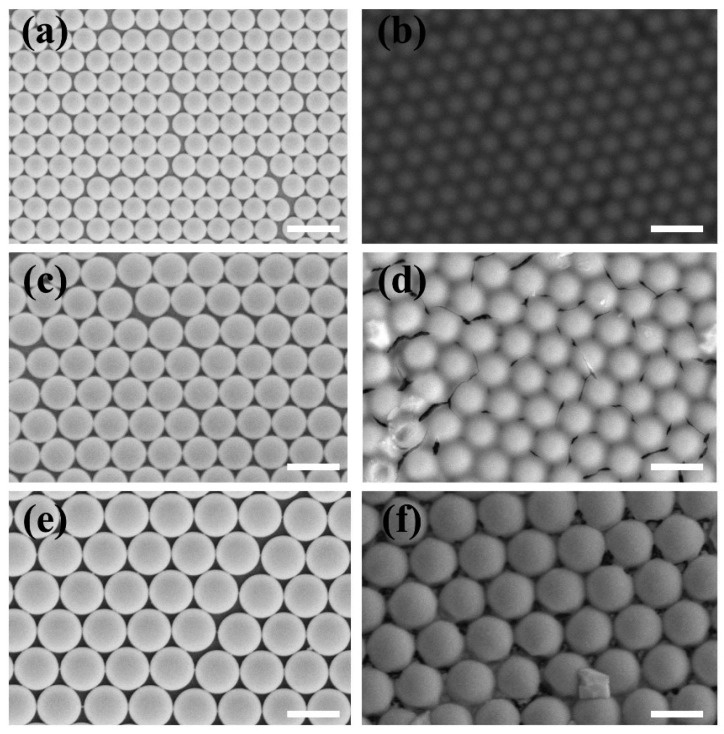
Scanning electron microscopy (SEM) images of PS arrays and CdSe/CdS QDs film coated on Au nanoshell arrays: (**a**) 500 nm PS arrays; (**c**) 750 nm PS arrays; (**e**) 1000 nm PS arrays; (**b**,**d**,**f**) CdSe/CdS QDs film coated on Au nanoshell arrays with 500 nm, 750 nm, and 1000 nm, respectively. Scale bar is 1 um.

**Figure 3 materials-12-00362-f003:**
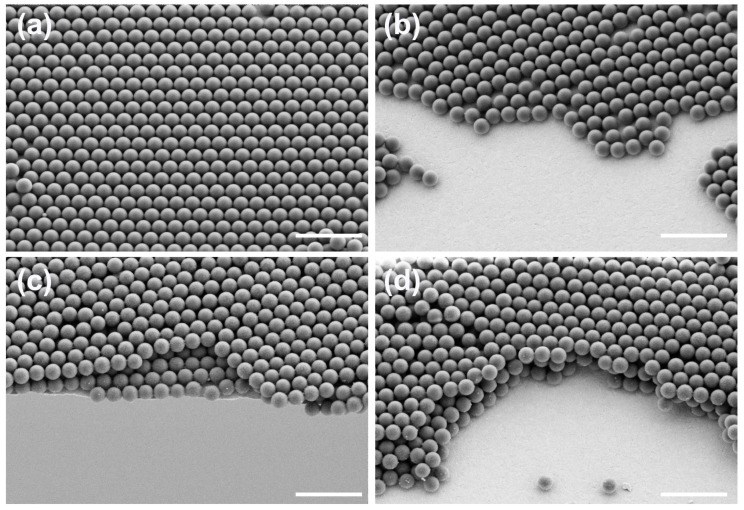
SEM of 500 nm PS arrays with different layers: (**a**) the top view of 500nm PS arrays; (**b**) the tilting 30° cross-sectional view of monolayer 500 nm PS arrays; (**c**) the tilting 30° cross-sectional view of double-layer 500 nm PS arrays; (**d**) the tilting 30° cross-sectional view of triple-layer 500 nm PS arrays, respectively. Scale bar is 2 um.

**Figure 4 materials-12-00362-f004:**
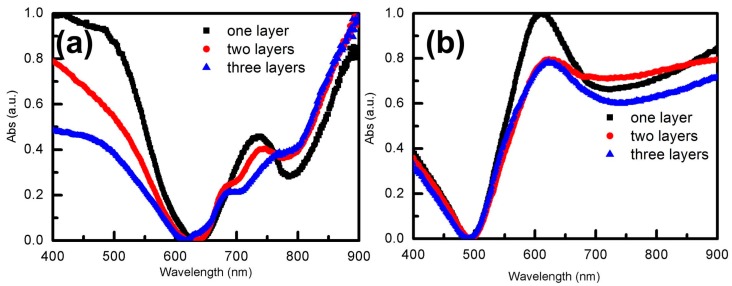
The absorbance spectra of Au shell arrays with the diameter of 500 nm: (**a**) absorbance spectra of Au shell arrays, with different layers of PS arrays, before annealing; (**b**) absorbance spectra of Au shell arrays, with different layers of PS arrays, after annealing.

**Figure 5 materials-12-00362-f005:**
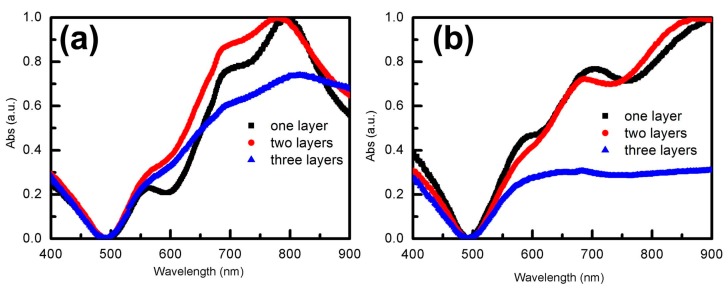
The absorbance spectra of Au shell arrays: (**a**) absorbance spectra of 750 nm Au shell arrays with different layers; (**b**) absorbance spectra of 1000 nm Au shell arrays with different layers.

**Figure 6 materials-12-00362-f006:**
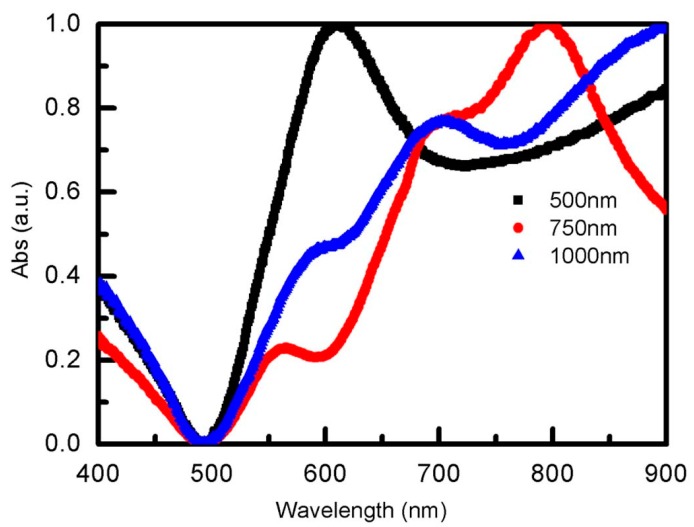
Absorption spectra of Au NS arrays with the diameters of 500, 750, and 1000 nm.

**Figure 7 materials-12-00362-f007:**
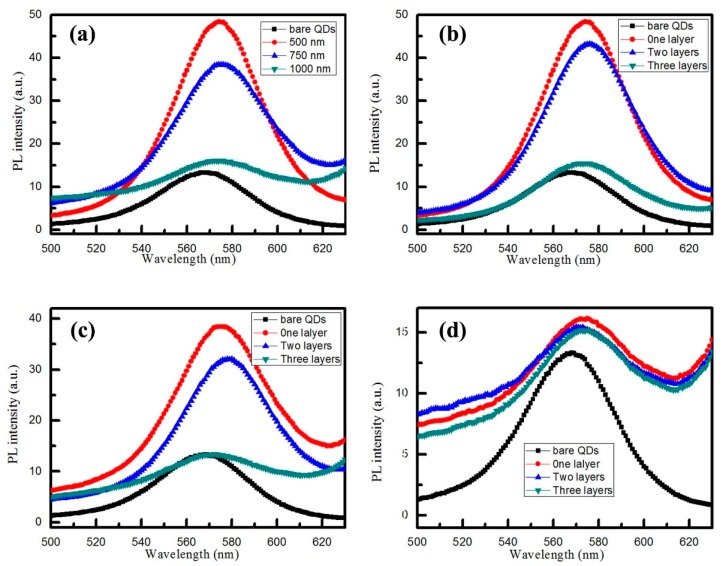
Room temperature PL spectra of bare CdSe/CdS QDs and CdSe/CdS QDs on Au NS arrays with different diameters: (**a**) bare CdSe/CdS QDs and CdSe/CdS QDs on different diameter—500, 750, and 1000 nm—Au NS arrays; (**b**) bare CdSe/CdS QDs and CdSe/CdS QDs on Au NS arrays with different layers of 500 nm PS; (**c**) bare CdSe/CdS QDs and CdSe/CdS QDs on Au NS arrays with different layers of 750 nm PS; (**d**) bare CdSe/CdS QDs and CdSe/CdS QDs on Au NS arrays with different layers of 1000 nm PS. The excitation wavelength was 325 nm.

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
