# Peer review of "Surface Plasmon-Enhanced Luminescence of CdSe/CdS Quantum Dots Film Based on Au Nanoshell Arrays"

_materials, 2019, doi:10.3390/ma12030362_

Reviewer 1 Report

Luo et al. present a study with CdSe/CdS QDs deposited on gold, which itself is nanostructured based on a template. Here they claim to see enhanced PL from the QDs when the plasmonic resonance of the gold matches the emission wavelength. This claim seems quite counter-intuitive as the gold should actually increase reabsorbance of the emitted light. On the other hand, matching the plasmonic resonance with the CdSe/CdS absorption would make more sense. In the abstract the authors state that the mechanisms is explained in the paper, however there seems to be little discussion about mechanisms, simply observations and correlations.

Given that QDs typically have strong absorption, and often high PLQY (depending on the capping ligand) i think it would be beneficial to the reader to explain why a plasmonic structure would be used to enhance these properties further. Typically (at least in photovoltaics) the primary use of plasmonic structures is with materials that have a low extinction coefficient, where simply making that layer thicker leads to other issues, such as recombination.

Making quantitative statements about photoluminescence can be challenging. Here the authors appear not to use a quantum efficiency technique, relying on just the intensity - presumably in close to identical conditions. Can the authors clarify if the 20 uL of QD *dispersion* was contained entirely within the film and whether it covered this homogeneously? It seems implausible that the CdSe/CdS QD would form a distinct film on top f the Au shell as shown in Figure 1, but would probably fill in the space. Also, what was the size of the film?

Some other issues which need to be addressed are listed below:

* In figure 1, panels g, h & i, as wel as j, k & l are identical. If this is correct, perhaps it would make sense to redraw so that each of d, e & f all point to the same subsequent panel.

* It seems a little odd for the discussion around the gold shell absorption to focus on transmission. I believe the transmission spectra are not necessary (these could be moved to SI?).

* It seems unusual that the 20 nm thick films would have *zero* absorbance at 500 nm. Can the authors explain this.

* The authors use a 325nm laser to excite the materials however there is no data presented on the absorbance of the gold shells at this wavelength, nor of the CdSe/CdS QDs. This is absolutely essential in order to establish the mechanism of increase PL. It could potentially even be reflection of the laser of this gold layer leading to increased absorption (the texture meaning it will scatter at different angles, increasing the path length through QDs and increasing the likelihood of absorption.

* A number of typographic errors have been repeated across several figures (eg "ttree layers", "One lalyer")

* The authors state that "The results indicated that Au film coated PS sphere could be completely decomposed by annealing at 300C". I suspect they mean that the PS is entirely removed, not the gold. Also, how did they check this? was there any residual carbon?

Author Response

Dear Ms. Sybil Zhang:

Thank you very much for your attention and the referee’s evaluation and comments on our paper “Surface Plasmon-Enhanced Luminescence of CdSe/CdS Quantum Dots Film Based on Au Nanoshell Arrays”. Manuscript ID: materials-410369. We have revised the manuscript according to your kind advices and referee’s detailed suggestions. Enclosed please find the responses to the referees. We sincerely hope this manuscript will be finally acceptable to be published on Materials. Thank you very much for all your help and looking forward to hearing from you soon.

Best regards

 Sincerely yours

 Wei-Guo Yan

Reviewer 2 Report

The aim of the study was to improve the fluorescent properties of CdSe / CdS QD core-shell type using nanoscale Au arrays fabricated based on polystyrene spheres (PS). The procedures of PS self-assemble with different sizes were the same as those in previously published works of authors. The variation of morphology and composition of the samples were investigated using scanning electron microscope. Optical properties of QDs on Au NPs surface were characterized by transmittance and absorption spectroscopy and photoluminescence measurements. However, a number of comments and questions arose to the discussion of the research results, their presentation and conclusion.

1. It is enough to show a schematic diagram of the technological process for a single-layer structure, since the technological process of obtaining multilayer structures differs only in the number of layers. Figure 1 should be changed.

2. To observe surface plasmon resonance in metal structures, absorption spectra are used. The transmittance spectra in Figures 4, 5, and 6 duplicate the absorption spectra. In this connection, it is necessary to revise the text.

3. The red shift and absorption rise of Au NS plasmon peak clearly sees in figures 4 (d) as result of annealing. However authors do not give comments on this. What is the nature of such significant changes in the spectrum?

4. What are the highlights obtained in this work in comparison with the previously known ones?

Author Response

Dear Ms. Sybil Zhang:

Thank you very much for your attention and the referee’s evaluation and comments on our paper “Surface Plasmon-Enhanced Luminescence of CdSe/CdS Quantum Dots Film Based on Au Nanoshell Arrays”. Manuscript ID: materials-410369. We have revised the manuscript according to your kind advices and referee’s detailed suggestions. Enclosed please find the responses to the referees. We sincerely hope this manuscript will be finally acceptable to be published on Materials. Thank you very much for all your help and looking forward to hearing from you soon.

Best regards

 Sincerely yours

 Wei-Guo Yan

Round  2

Reviewer 1 Report

Luo et al. have made a number of changes to their manuscript. Unfortunately, some of their responses appear to miss the point(s) of my original criticism. As such i cannot recommend publication at this point.
Going through the points from last time:
1 - The authors claim low PL from QDs in films, however (1) they do not provide any numbers to back this up - at the least, some literature values for similar materials should be included & (2) there is still no explanation of the mechanism by which plasmonic structures would increase the PLQY. My understanding is that this would only increase the absorbance, which could also be achieved by making thicker films. The authors should include an explanation of the mechanism they mention (enhancing emission in competition with recombination?), including good references to the literature. NB Generally PLQY is an internal quantum yield value, ie the number of photons emitted divided by the number absorbed, rather than and EQE, which would be photons emitted divided by the number shone on the sample.
2 - With the films being hydrophilic, can the authors confirm whether some of the QD dispersion also escaped out from the film? or was it all contained within the 1.5 x 37.5px section? Did they quantify that the same number of QDs were present for each of the samples (including the one without any PS bead layers?
3 - Thank you for amending this, it is much easier to follow now.
4 - The discussion focuses on absorption valleys, which are the same as transmission peaks. The reason I asked about this that my understanding of the process is that it is related to enhanced absorption of particular wavelengths (plasmonic absorption), rather than wavelengths where absorption is low?
5 - I am unclear what the authors mean by 'intrinsic absorption of Au' at 500nm, and why this is zero? How were these measurements made? It almost sounds as if a 20 nm film of gold on a glass substrate used as the reference for baselining the photospectrometer?
6 - the reason i asked about absorption at 325nm is that this is where you excite your QDs in the PL experiments. Changes in the absorption (plasmonic or otherwise) of gold layers at this wavelength will affect how much light the QDs absorb. Also, in Fig 7, the value of 350 nm is stated for excitation.
7 - thank you for addressing these typos
8 - please add a reference and a short sentence to the manuscript to explain your assumption about the complete removal of PS at 300C.
In addition, when looking at Figure 7, it seems a little odd that there is such a substantial PL signal away from the peak, especially when either 1000 nm PS beads are used, or three layer films are made. While the uptick beyond 620 nm may be attributed to harmonic transmission though monochromating optics of the PL system, this doesn't explain the strong response between 500 and 550 nm. Can the authors explain this and why this phenomenon is not seen as much with other experimental conditions?

Author Response

Dear Ms. Sybil Zhang:

Thanks for the Reviewer’s excellent evaluation and kind suggestion about our paper titled “Surface Plasmon-Enhanced Luminescence of CdSe/CdS Quantum Dots Film Based on Au Nanoshell Arrays” Manuscript ID: materials-410369. The attached document1 are the cover letter for reviewers. In addition, the cover letter for reviewers is submitted in the submission system.

We greatly appreciate the efficient, professional and rapid processing of our paper by your team. If there is anything else we should do, please do not hesitate to let us know.

Thank you and best regards.

Thank you and best regards.

Yours sincerely,

Wei-Guo Yan

Reviewer 2 Report

The authors took into account the comments and recommendations, and made the corresponding corrections in the text.

Author Response

Thank you for your valuable comments.

Round  3

Reviewer 1 Report

Again Luo et al. have made changes to their manuscript, however some previously raised questions/criticisms have not yet been fully addressed:

* re: comment 1 - Thank you for finally including some references, although not all of these are pertinent to the work here. The authors should provide more detail on the mechanism by which plasmonic resonance is enhances PL emission, based on matching of plasmonic resonance and QD emission spectra. Intuitively enhanced (plasmonic) absorption would  simply mean that more emitted light (from the QD) would be re-absorbed and not be seen. Digging into the supplied references, and then into the papers *they* reference, it looks that the mechanism indeed is an enhancement of the emission quantum yield. Ideally the authors should calculate and report PLQY for each of their samples. At the very least they need to convince the reader that their samples are directly suitable for semi-quantitative comparison. The authors still need to  measure and report the absorption(s) at 325 nm (where they excite their QDs) to verify that the enhancement is due to the proposed phenomenon, rather than other optical effects.

* re: comment 5 - the authors have not added any additional experimental detail as requested. Please clarify what you mean by "intrinsic absorption (adsorption?)". It appears that the patterned gold layer on a glass substrate has the same absorbance as the glass alone at this wavelength (500 nm). Is this correct? A quick lit search shows others measuring the absorbance of 20 nm gold films (eg https://arxiv.org/ftp/arxiv/papers/1409/1409.7338.pdf) with an absorption in excess of 70% of incident light, even at the lowest point.

* re: comment 6 - The fact remains that the authors have not presented absorption spectra at the excitation wavelength used in PL experiments. There is no way for the reader to know how much difference there is in the light being harvested between samples.

* re: comment 8(b) - the 'weak' PL here is still more intense than the sample without the plasmonic resonator, which does not display the > 620 nm uptick.

Author Response

Dear Reviewer:

Thanks very much for your kind letter about our paper titled “Surface Plasmon-Enhanced Luminescence of CdSe/CdS Quantum Dots Film Based on Au Nanoshell Arrays” Manuscript ID: materials-410369. The attached document1 is revised manuscript and document2 are the cover letters for reviewers. 

We greatly appreciate the efficient, professional and rapid processing of our paper by your team. If there is anything else we should do, please do not hesitate to let us know.

Thank you and best regards.

Thank you and best regards.

Yours sincerely,

Wei-Guo Yan
